chemical biology/environmental science/environmental engineering

lead-resistant bacteria, biosorption, bioimmobilization, wastewater

**Author for correspondence:**
Weichuan Qiao
e-mail: hgqwc@njfu.edu.cn

# Bioimmobilization of lead by *Bacillus subtilis* X3 biomass isolated from lead mine soil under promotion of multiple adsorption mechanisms

Weichuan Qiao, Yunhao Zhang, Hao Xia, Yang Luo, Si Liu, Shiyu Wang and Weihan Wang

Department of Environmental Engineering, College of Biology and the Environment, Nanjing Forestry University, Nanjing 210037, People's Republic of China

iD WQ, 0000-0001-9721-9364

In this study, a lead-resistant bacterium, *Bacillus subtilis* X3, was used to prepare a lead bioadsorbent for immobilization and removal of lead in lead solution. The lead shot precipitate was analysed by scanning electron microscopy combined with energy dispersive X-ray fluorescence microscopy, Fourier transform infrared spectroscopy, X-ray diffraction and X-ray photoelectron spectroscopy. The adsorbed lead was mainly mineralized to form $Pb_5(PO_4)_3OH$, $Pb_{10}(PO_4)_6(OH)_2$ and $Pb_5(PO_4)_3Cl$; however, other mechanisms that can also promote the mineralization of lead should not be ignored. For example, $Na^+$ and $Ca^{2+}$ on the cell wall surface were exchanged with $Pb^{2+}$ in solution, which confirmed that the ion-exchange process occurred before mineralization. Moreover, adsorption bridging caused by extracellular polymeric substances also accelerated the further aggregation of lead, and the biomass was encapsulated by lead gradually. Hydroxyl, carbonyl, carboxyl and amine groups were not observed in lead mineral crystals, but the complexation between lead and these groups still benefited the mineralization of lead. The valence of Pb(II) was not changed after mineralization, which indicated that the biosorption process was not a redox reaction. Finally, biosorption occurred on the outer surface of the cell, but its specific surface area was relatively small, limiting the amount and efficiency of biosorption.

## 1. Introduction

Heavy metal pollution is a global environmental problem and has received a great deal of attention. Unlike traditional pollutants,

heavy metals are generally stable and not biodegradable, although their concentrations are quite low [1,2]. Mining, electroplating, metal processing, textiles and the battery manufacturing industry are the main sources of lead contamination [3]. Once lead is discharged into an aqueous environment, it will be concentrated in fish, vegetables and other foods, which then impacts humans via the food chain, resulting in strong toxicological effects on the human heart, liver, kidneys, brain and reproductive systems, especially in children [4–8]. Accordingly, emissions of lead must be controlled.

Traditional methods for removing lead from aqueous environments include chemical precipitation, solvent extraction, ion exchange, coagulation, air flotation and activated carbon adsorption [9–11]. When compared with low efficiency, high costs and the environmentally unfriendly defects of these methods, bioremediation is regarded as one of the most cost-effective methods for removal or recovery of heavy metals from wastewater [12]. Bioremediation of lead in the environment reduces the water solubility of lead, and then reduces the uptake of lead by plants or the formation of lead precipitate that can be removed from the water. Many studies have reported that a wide variety of microorganisms have high tolerance and biosorption abilities for lead, including algae, fungi, yeasts and bacteria [13–16].

Lead-immobilization microorganisms can be introduced into contaminated water or soil, after which they grow and immobilize lead through complexation, coordination, physical adsorption, chelation, ion exchange, inorganic precipitation or some combination of these processes, thereby reducing the hazard posed by lead in the environment [17]. The nutrients in soil are rich and suitable for microbial growth; therefore, the introduction of lead-immobilization microbes is an effective method of remediating contaminated soil. However, when microorganisms are introduced into lead-contaminated water, they perhaps grow slowly because there are usually not enough nutrients, but many toxic substances. Therefore, the effects of remediation of using live biomass can not be satisfied.

Biosorption is the binding of microbial biomass to heavy metals by physical or (and) chemical methods [18]. The biosorption of lead is mainly achieved through cell wall functional groups, although it can also occur via extracellular polymeric substances (EPS) secreted from microbes [19,20]. The ability to remove and adsorb metal is also significantly higher than that of intracellular substances [21]. The cell wall is negatively charged as a result of functional groups, including phosphate, carboxyl, carbonyl, sulfhydryl and hydroxyl groups, which can bind lead to form insoluble substances. Cations such as $Na^+$ and $K^+$ can exchange $Pb^{2+}$ on the outer layer [22]. Moreover, the cell wall is composed of organic macromolecules, such as polysaccharides, polypeptides and proteins, which can adsorb lead via electrostatic forces, Van der Waals' forces, covalent bonds or ion exchange [23,24]. Moreover, the microfibrous porous structure of the cell wall means that lead will be deposited on the surface of the cell wall or embedded in the cell wall before entering the cell.

Many researchers have used microbial biomass to prepare adsorbents to reduce the effects of adverse environmental conditions on microbial growth [4,25,26]. The functional groups and ions on cell walls can still play roles in adsorption despite the biomass being dead. However, the characteristics of dead biomass are invariable, which is different from live biomass. Although we know that a combination of different processes is the cause of lead biosorption, it is not clear which process is the key mechanism and what the relationship between these mechanisms is. In this study, we isolated and identified a lead-resistant *Bacillus* sp. from lead-polluted soil in a lead mine plant, and then analysed the characteristics of lead shots produced during the biosorption process. The lead bioimmobilization mechanisms of the microbial biomass in the aqueous environment were also evaluated.

# 2. Material and methods

## 2.1. Isolation and identification of microorganism

The lead-resistant bacterial strain was isolated from the lead-contaminated soil of a lead mine in Nanjing, Jiangsu Province, China. To enrich the bacteria, 10 g of soil was added into 100 ml Luria–Bertani (LB) basal medium and then incubated at 150 rpm and 37°C for 48 h. Next, 1 ml of the above culture was added into 100 ml basal medium containing 400 mg l$^{-1}$ Pb(NO$_3$)$_2$ and cultured under the same conditions described above. The isolates were then purified by the dilution-plate method [27]. Individual bacterial colonies were isolated on LB medium containing 400 mg l$^{-1}$ Pb(NO$_3$)$_2$. The formation of black lead shot precipitation and lead reduced in the culture indicated that the bacterial colony could immobilize lead. The concentration of lead in solution was detected by flame atomic adsorption spectrometry (TAS-900, PGENERAL).

Genomic DNA of strain X3 was extracted using a Bacterial Genomic DNA Extraction Kit (Sangon, China). The 16S rRNA gene was amplified using the primers f1 (5′-AGTTTGATCMTGGCTCAG-3′) and r1 (5′-GGTTACCTTGTTACGACTT-3′) by polymerase chain reaction, after which the fragments were purified using a gel extraction kit (Sangon, China), and the sequencing of the 16S rRNA gene was conducted by Sangon (Shanghai, China). The BLAST programme (http://blast.ncbi.nlm.nih.gov/Blast.cgi) was used for a sequence similarity search with the standard programme by default. Multiple sequence alignment and data analysis were conducted using the software package MEGA v. 5.1, and a phylogenetic tree was constructed using the neighbour-joining method.

## 2.2. Bacterial biomass adsorption preparation

The isolated and purified strain X3 was inoculated in LB medium and cultured at 37°C and 150 rpm for 24 h. The cells were then harvested by centrifugation for 10 min at 5000$g$, after which the biomass was rinsed with sterile deionized water three times to remove the culture solution. Finally, the obtained cells were freeze-dried using a lyophilizer.

## 2.3. Lead adsorption experiments

The prepared biomass adsorbent was placed in 100 ml solution containing $Pb(NO_3)_2$ (200, 400, 600, 800, 1000, 1200, 1400 mg l$^{-1}$ of $Pb^{2+}$). The pH of the solution was adjusted to 1–5 by 0.1 M NaOH solution and 0.1 M $HNO_3$ solution. Next, each solution was amended with biomass (0.01, 0.02, 0.03, 0.04, 0.05, 0.06, 0.07 and 0.08 g). The reaction system was then placed in a shaker at 25°C at 150 rpm for half an hour, after which it was allowed to stand for 1–30 min. Following centrifugation at 8000 rpm for 5 min, the concentration of lead in the supernatant was detected by flame atomic adsorption spectrometry. Each experiment was performed with three biological and technical replicates. The 95% confidence interval ($p < 0.05$) was set as the significance threshold. The lead shot precipitate was collected by rinsing three times using sterile deionized water. Finally, the adsorption capacity of the biomass adsorbent was calculated by the following formula:

$$q_e = \frac{C_0 - C_e}{M} V,  \tag{2.1}$$

where $C_0$ and $C_e$ were the initial and final concentrations of lead (mg l$^{-1}$), respectively, and $V$ and $M$ were the volume of solution (l) and the weight of the biomass (g), respectively.

## 2.4. Characterization of lead shot

The collected lead shot was dried at 105°C for 2 h, then used for analysis. The surface morphological and elemental components of the lead shot could be analysed by scanning electron microscopy combined with energy dispersive X-ray fluorescence microscopy (SEM–EDS, Shimadzu, UK). The SEM spectra enabled the direct observation of changes in the surface structures of the bacteria [5]. Lyophilized samples of the bacteria before and after contact with $Pb^{2+}$ were used for X-ray diffraction (XRD) analysis (Rigaku, Japan). Diffraction patterns were collected at angles ranging from 15° to 65° with a 0.02° step-length. X-ray photoelectron spectroscopy (XPS, Shimadzu) was used to characterize the surface chemical compositions of the lyophilized samples [28], while the surface functional groups of the bacteria were analysed by Fourier infrared spectroscopy [5], after diluting the adsorbent to 5% in KBr and casting the samples in disks (Brooke, Germany).

# 3. Results

## 3.1. Isolation and identification of bacteria

Seven different isolates exhibiting lead resistance were obtained from the lead-contaminated soil at a lead mine plant. One isolate, strain X3, was selected for further study because of its high lead tolerance and removal rate. The 16S rRNA sequence of strain X3 was amplified and sequenced, after which the sequence was submitted to GenBank under accession number KX966417. The results of a BLAST search indicated that this strain shared 99% similarity to *Bacillus subtilis*. A phylogenetic tree demonstrated that strain X3 was closely related to other *Bacillus* sp., but was most closely related to

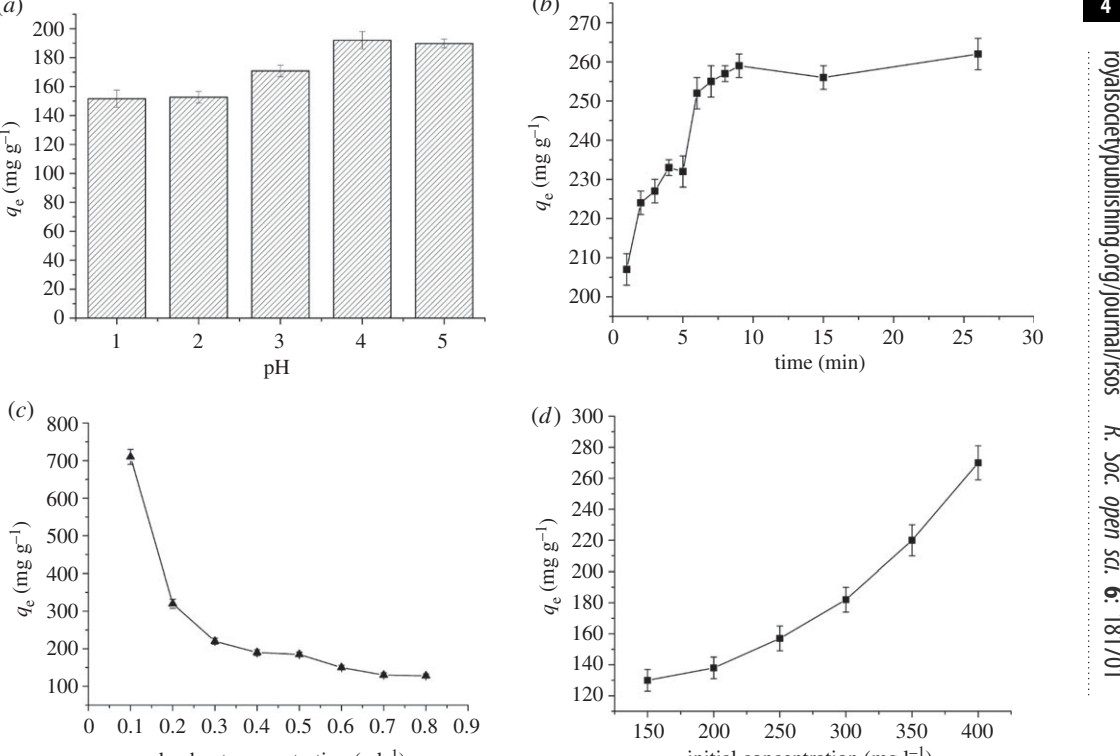

**Figure 1.** Effects of different influencing factors on adsorption of lead by bacterial biosorbent.

*Bacillus subtilis* (NC000964.3) (electronic supplementary material, figure S1). Finally, strain X3 was identified as *B. subtilis* X3 based on the results of the morphological observations and the 16S rRNA sequence analysis. The growth of *B. subtilis* X3 on plates with different lead concentrations showed that the maximum amount of lead *B. subtilis* X3 could tolerate was 2000 mg l$^{-1}$.

## 3.2. Different factors in biosorption

To investigate the effects of different pH on the adsorption of biomass, the pH of solution was adjusted from 7.0 to 1.0. When the pH was 6.0, some lead shot precipitate appeared. The lead removal at pH values of 1–5 are shown in figure 1a. The adsorption capacity increased with the increase in pH, eventually reaching the maximum value of 192.05 mg g$^{-1}$.

Contact time is an important factor influencing the removal rate of lead biosorption. Batch experiments were conducted for different reaction times to determine the optimal contact time for the biomass of *B. subtilis* X3 (figure 1b). Briefly, 0.11 g of biomass was added into 50 ml of solution with 400 mg l$^{-1}$ Pb(NO$_3$)$_2$. The amount of bioadsorbed lead increased with increased contact time, and reached a maximum of 260 mg g$^{-1}$ at 10 min.

The dosage of biomass was also an important factor influencing the adsorption of lead. To ensure its effects on lead adsorption, varying amounts were introduced into lead solution. As shown in figure 1c, the adsorption capacity of the biomass was 703.25, 321.875, 228.08, 188.62, 180.9, 144.83, 130.46 and 133.31 mg g$^{-1}$. These results showed that the adsorption capacity of the adsorbent decreased as the biomass increased.

To investigate the effects of initial lead concentration on biomass adsorption, the lead concentration after adsorption was determined under the initial concentration of 150–400 mg l$^{-1}$. As shown in figure 1d, the adsorption amount increased with the initial lead concentration, and the adsorption capacities were 129.8, 137.9, 163.55, 194.05, 218.95 and 283.75 mg g$^{-1}$. These results showed that the amount of lead adsorbed increased with the initial lead concentration. High concentrations can increase the mass transfer power of lead and overcome the obstacle of mass transfer between the solid and liquid phase of lead. When the concentration of lead in the solution was 350–400 mg l$^{-1}$, the effects of cell adsorption increased, indicating that there is a more rapid and effective mechanism of adsorption in solutions containing high concentrations of lead.

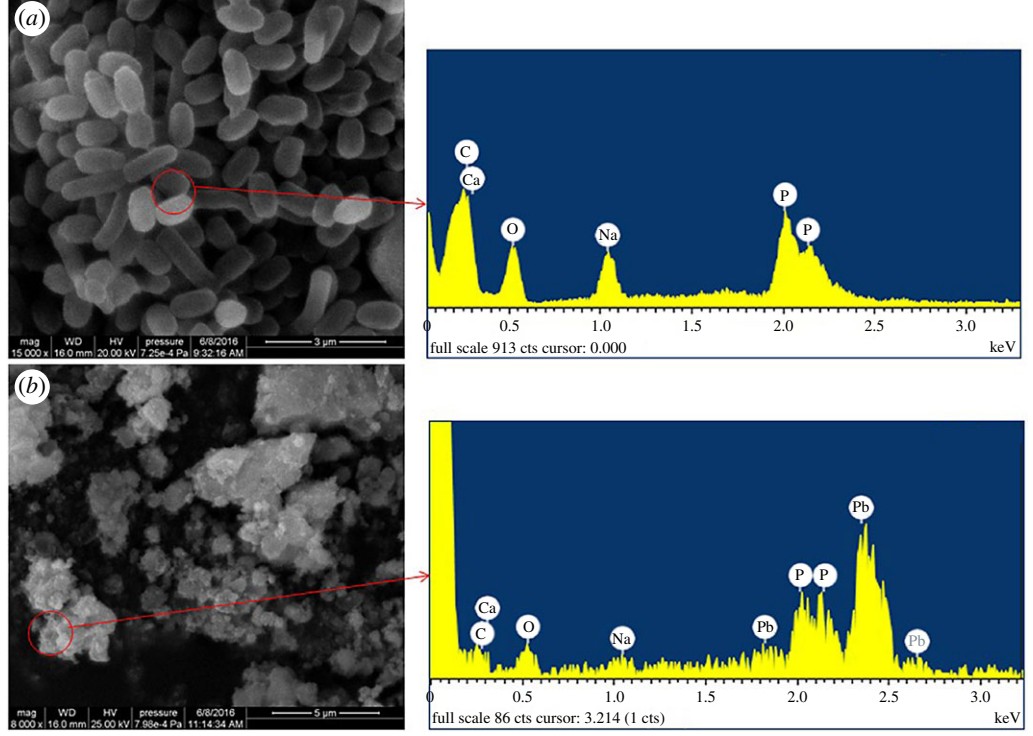

**Figure 2.** SEM−EDS photographs of the lead shot precipitate and *B. subtilis* X3.

## 3.3. Characteristics of lead shot

### 3.3.1. Observation of lead shot by SEM−EDS

When the biomass of *B. subtilis* X3 was added into lead solution, lead shots were formed, indicating that lead in the solution was immobilized and removed. As shown in figure 2, the lead shot was characterized by SEM−EDS. The cells were rod shaped, with a smooth surface and very small diameter prior to lead adsorption (figure 2*a*). However, when the lead shot formed, there was no complete visible bacterial surface and some of the aggregated lead particles were wrapped together. A large amount of lead particles gathered around the adsorbed cells.

The chemical components of the lead shots were analysed using EDS images. As shown in table 1, the percentage of phosphorus, calcium and lead increased, while that of sodium decreased from 3.16 to 1.66% when compared with the control. These results showed that ion exchange occurred between lead, calcium and sodium. Not only was lead-hydroxyapatite formed, but also $Ca_3(PO4)_2$. Moreover, the decrease in carbon and oxygen ion showed that the lead shot had covered the cell biomass.

### 3.3.2. XPS of lead shot

The chemical composition of the cell surface and identification of the oxidized state of the lead samples were evaluated by XPS analysis (figure 3). The results of XPS indicated that the core level peaks were C-1s (284.6 eV), Pb-4f (138.40 eV) and Pb-4f5/2 (142.70 eV). The C-1s peak fluctuated around 284.1 eV, indicating that the C (C, H) or amino acid side chain functional groups were involved in the adsorption process and that carboxylate or ester C=O functional groups were at 289.0 eV during this process. Pb-4f7/2 and Pb-4f5/2 peaks were observed at 138.40 eV and 142.70 eV, respectively. The corresponding spin orbital division of the peak was 5.01 eV, indicating that the lead adsorbed on the outer surface by the bacterial adsorbent was still in the divalent valence state.

### 3.3.3. XRD of lead shot

Compounds that undergo lead transition during adsorption were identified by XRD spectral analysis. As shown in figure 4, a new adsorption peak appeared at 30° after adsorption. Analysis using the MID Jade

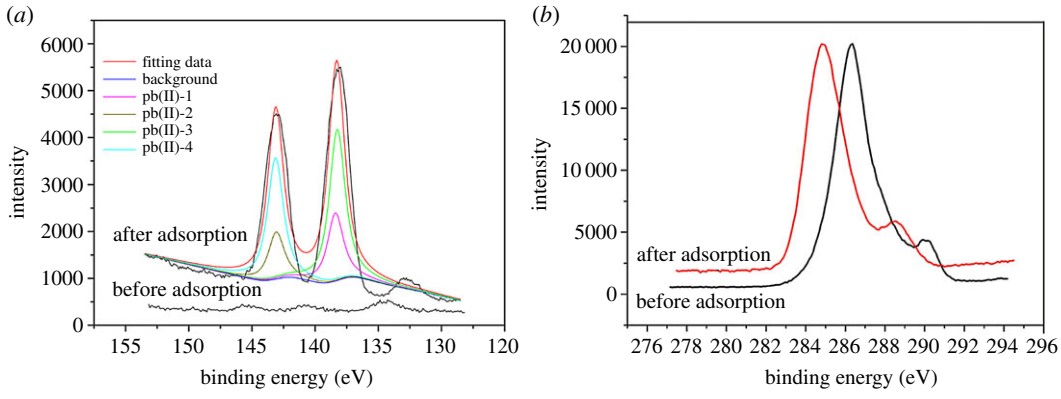

**Figure 3.** XPS spectra of the lead shot precipitate and *B. subtilis* X3 biomass, (*a*) Pb-4f spectrum; (*b*) C-1s spectrum.

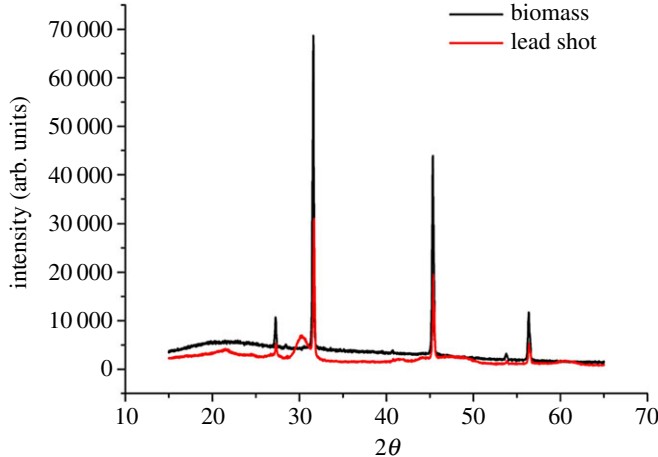

**Figure 4.** XRD spectra of the lead shot precipitate and *B. subtilis* X3 biomass.

**Table 1.** Elemental analysis of the surface of the adsorbent before and after adsorption.

| element | control | | lead precipitate | |
| --- | --- | --- | --- | --- |
| | weight % | atomic % | weight % | atomic % |
| C | 49.42 | 58.69 | 23.05 | 46.43 |
| O | 40.04 | 35.70 | 27.81 | 42.07 |
| Na | 5.09 | 3.16 | 1.58 | 1.66 |
| P | 4.93 | 2.27 | 5.76 | 4.50 |
| Ca | 0.52 | 0.18 | 0.93 | 0.56 |
| Pb | — | — | 40.88 | 4.77 |
| totals | 100.00 | — | 100.00 | — |

software analysis showed that $Pb^{2+}$ was transferred to $Pb_5(PO_4)_3$, $Pb_{10}(PO_4)_6(OH)_2$ and $Pb_5(PO_4)_2Cl$ during the adsorption process.

### 3.3.4. FT-IR of lead shot

The results of the Fourier transform infrared spectroscopy (FT-IR) can be used to determine functional groups on the outer surface of the lead shot and their sources. Many functional groups on the outer surface of bacteria can chelate or complex lead, such as hydroxyl, carboxyl and phosphate groups. The general distribution of the various major groups on the lead shot in the infrared adsorption spectrum is shown in figure 5. The results showed that there was a broad and strong adsorption peak in the

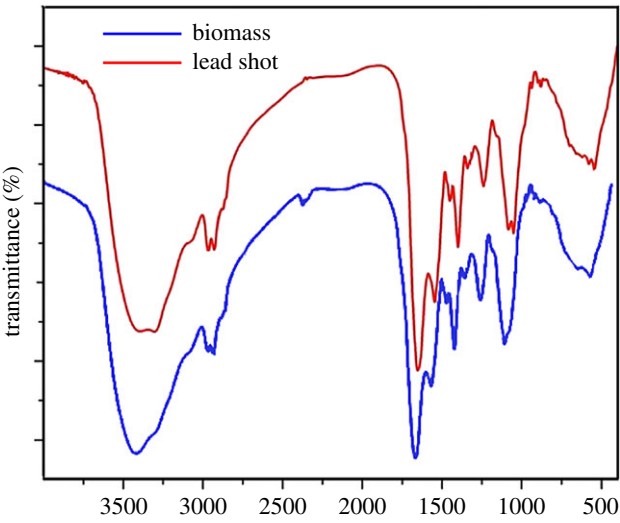

**Figure 5.** FI-IR spectra of the lead shot precipitate and *B. subtilis* X3 biomass.

**Table 2.** Comparison of adsorption capacity of different microorganisms.

| microbes | removal capacity (mg g$^{-1}$) | references |
|---|---|---|
| *Bacillus* sp. | 92.27 ± 1.17 | Tunali *et al.* [32] |
| *Bacillus cereus* | 36.71 | Pan *et al.* [33] |
| *Pseudomonas putida* | 270.44 | Uslu *et al.* [34] |
| *Chlorella vulgaris* | 178.5 | Edris *et al.* [26] |
| *Ecklonia radiata* ER95Ca | 269.3 | Matheickal *et al.* [35] |
| *Actinomycete strain* 723 | 116.0 | Karaduman *et al.* [36] |
| *Streptomyces rimosus* | 135 | Selatnia *et al.* [37] |
| *Streptomyces noursei* | 36.5 | Mattuschka *et al.* [38] |
| *Aspergillus flavus* | 13.46 ± 0.99 | Akar *et al.* [14] |
| *Neurospora crassa* | 49.06 | Kiran *et al.* [39] |
| *Botrytis cinerea* | 107.10 ± 1.87 | Akar *et al.* [29] |
| *Saccharomyces cerevisiae* | 270.3 | Ozer *et al.* [40] |
| *Penicillium sp.* MRF-1 | 72.5 | Velmurugan *et al.* [41] |
| *Bacillus subtilis* X3 | 590.49 | this study |

region of 3500–3300 cm$^{-1}$, which was the stretching peak of O–H and N–H derived from carboxylic acid, amino acid or alcohol [29]. The adsorption peak at 2928 cm$^{-1}$ can be assigned to C–H stretching. Obvious changes between biomass and the lead shot at 2361 cm$^{-1}$ reflecting the C≡C stretching vibration peak were observed. The adsorption peak at 1650 cm$^{-1}$ was indicative of C=O group [30]. Additionally, a P=O stretching vibration peak at 1400 cm$^{-1}$ derived from the phosphoric acid group was observed, as was a C–N stretching vibration peak of the saccharide at 1236 cm$^{-1}$. Finally, a stretching vibration peak at 1050 cm$^{-1}$ originated from the OH of alcohol or C–N groups [31]. These results showed that the adsorption process was mainly because of the adsorption of functional groups on the outer surface of the biomass, which was confirmed by the experimental results of FT-IR.

## 4. Discussion

In this study, lead-resistant *B. subtilis* X3 was isolated from the soil of a lead mine plant, formed into a biomass adsorbent and applied to the removal of lead in solution. When compared with similar studies (table 2), the biomass generated here has good potential for practical application.

Adsorption of lead in solution by biomass is influenced by many factors, with solution pH being one of the most important. In this study, the adsorption capacity of *B. subtilis* X3 biomass reached the maximum amount when the pH was 4, which was similar to the results of previous reports [42]. Ren *et al.* [5] pointed out that pH is one of the most important parameters affecting the solubility of metal ions and functional groups on the cell wall of biomass. A functional group such as a carboxylate on the bacterial cell wall is protonated below 4.0, and Pb(II) is weak in competition with hydrogen ions at the adsorption site on the surface of the cell. Therefore, the lower Pb(II) absorption capacity was caused by more hydrogen ions at a pH of 3.0–4.0 [5].

The initial lead concentration also affected the biosorption of lead. The amount of *Bacillus thuringiensis* 016 adsorbed increased with the increasing initial lead concentration [42], but in a study conducted by Ren *et al.* [5], the amount of lead biosorbed to *Bacillus* sp. PZ-1 decreased with increasing initial concentration. When compared with other microorganisms, the time required to reach biosorption equilibrium in the *B. subtilis* X3 biomass was obviously shorter.

Adsorption during biosorption occurs via ion exchange, complexation and biomineralization [43]. In previous studies, $Pb^{2+}$ was found to be transformed to $PbHPO_4$, $Pb_9(PO_4)_6$ or PbS [44,45]. In the present study, the products of $Pb^{2+}$ biomineralization were $Pb_5(PO_4)_3OH$, $Pb_{10}(PO_4)_6(OH)_2$ and $Pb_5(PO_4)_3Cl$. However, we found no evidence of PbS and $PbSiO_3$ formation [46]. We assumed that the formation of lead–phosphorus was determined by the species of microorganism, amount and kinds of functional groups on the cell surface and the solution composition. The immobilization of lead occurs via a complexation reaction between $Pb^{2+}$ in solution and functional groups on the cell surface that lead to formation of an insoluble substance. These functional groups include hydroxyl, carbonyl, carboxyl, amine and phosphoric acid groups [15]. In this study, we found the same groups on the lead shot upon FT-IR spectral analysis, but we did not find these functional groups upon XRD analysis of the lead mineral crystals. We assumed that complexation could help form the lead mineral, although the process did not play a major role. Furthermore, we found that some C≡C bonds bound lead, which was different from the results of other studies. However, we did not conclude from which substances the C≡C bond was derived.

Adsorption of $Pb^{2+}$ onto the biomass occurred via ion exchange with $Na^+$, $K^+$ and $Ca^{2+}$ in solution. In this study, the concentration of $Na^+$ decreased in solution, while it increased obviously in the lead precipitate (figure 2). The EPS secreted by microbes was an important factor influencing lead adsorption. Moreover, EPS was usually exported to solution to bind lead, although it could remain on the surface of microbes even though microbial biomass was obtained by centrifugation and freeze-drying. We assumed that adsorption bridging action promoted the immobilization of lead, and that this was contributed to by EPS on the surface of the microbial biomass. Electron transfer will not occur during binding of $Pb^{2+}$ and functional groups [47]. The valence of lead did not change, indicating that there was no electron transfer and that the lead–phosphorus mineral was formed by electrostatic incorporation.

# 5. Conclusion

In this study, *B. subtilis* X3 isolated from the soil in a lead mine plant was able to adsorb lead efficiently in solution. To investigate its adsorption mechanism, SEM–EDS, XRD, XPS and FT-IR methods were applied to analyse the characteristics of lead shot precipitate formed during adsorption. The analysis showed that the lead minerals formed were composed of $Pb_5(PO_4)_3OH$, $Pb_{10}(PO_4)_6(OH)_2$ and $Pb_5(PO_4)_3Cl$, which are the major immobilization mechanisms. During the mineralization of lead, ion exchange and adsorption bridging played important roles, indicating that they can be used to promote mineralization. The biosorption of lead mainly occurred on the outer surface, where lead can bind with hydroxyl, carbonyl, carboxyl, amine and phosphoric acid functional groups. We also found that C≡C bonds were involved in the binding of lead. These groups can accelerate the formation of lead mineral crystals, although they were not found in lead shots by XRD analysis.

Data accessibility. DNA sequences: Genbank accessions KX966417.

Authors' contributions. Y.Z. carried out isolation and identification of the lead-resistant bacteria, participated in data analysis; H.X. carried out the adsorption experiments of bacteria biomass, and the design of the study and drafted the manuscript; H.X. and S.L. carried out the acquisition of data, analysis and interpretation of data; Y.L., S.W. and W.W. carried out determination of lead shot characteristics; W.Q. conceived of the study, designed the study, coordinated the study, helped draft the manuscript and revised it critically for important intellectual content, and final approval of the version to be published. All authors gave final approval for publication. All authors agreed to

be accountable for all aspects of the work in ensuring that questions related to the accuracy or integrity of any part of the work are appropriately investigated and resolved.

Competing interests. We have no competing interests.

Funding. This work was supported by the Priority Academic Program Development of Jiangsu Higher Education Institutions (PAPD).

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
