## [Reviewer comments · Royal Society Open Science]

Review History

RSOS-181701.R0 (Original submission)

Review form: Reviewer 1

Is the manuscript scientifically sound in its present form?

No

Are the interpretations and conclusions justified by the results?

No

Is the language acceptable?

Yes

Is it clear how to access all supporting data?

No

Do you have any ethical concerns with this paper?

No

Have you any concerns about statistical analyses in this paper?

No

Recommendation?

Major revision is needed (please make suggestions in comments)

Comments to the Author(s)

In this paper, a lead-resistant bacterium, *Bacillus subtilis* X3, was used to prepare a lead bioadsorbent for immobilization and removal of lead in lead solution. Over the topic is interesting, however the paper is missing so much basic information. Major revision is needed if this paper can be accepted.

1. Page 3, line 24, The authors should explain why the adsorption capacity increased with the increase of pH, not just present a phenomenon.
2. Fig. 1(A) and (B), what do the abscissas represent?
3. Fig. 1(C), this figure should be revised and the title of abscissas should be revised to adsorbent concentration (g/L).
4. 4.3.3 XRD analysis, page 4, line 32-33, the authors stated that Pb^{2+} was transferred to $Pb_5(PO_4)_3$, $Pb_{10}(PO_4)_6(OH)_2$ and $Pb_5(PO_4)_2Cl$. However, there is no any analysis. You absolutely have to use citations to back up your statements.
5. The experimental conditions of Figure 1 should be provided in figure title.
6. The title of figure 3. XPS spectra of the lead shot precipitate and *B. subtilis*X3 biomass, (a) full spectrum; (b) Pb-4f spectrum; (c) C-1s spectrum. However, there is only one figure. Please explain.
7. The figures in page 9-11 and figures in page 12-16 are same.

Review form: Reviewer 2

Is the manuscript scientifically sound in its present form?

Yes

Are the interpretations and conclusions justified by the results?

Yes

Is the language acceptable?

Yes

Is it clear how to access all supporting data?

Yes

Do you have any ethical concerns with this paper?

No

Have you any concerns about statistical analyses in this paper?

No

Recommendation?

Accept with minor revision (please list in comments)

Comments to the Author(s)

This manuscript by Qiao, Zhang, and Luo is an overall well-written report on their study with lead bioimmobilization by *Bacillus*. The authors used several lines of evidence to show sorption of lead onto the biomass. There are only a few details in the manuscript that need to be edited/added in order to make the methods more clear.

Specific comments:

Page 1 Line 22 – the “*subtilis*” is not capitalized.

Page 2 Line 27 – What does ‘manually’ mean here with regards to genomic DNA extraction. Is there a protocol to reference? If not, some brief details on genomic DNA extraction need to be provided.

Page 2 Line 28 – What primers were used for the PCR of the 16S rRNA gene?

Page 2 line 43 – What is the form of lead here? Is it lead nitrate still? Be specific.

Page 2 line 59 – The q_e in the figures is in terms of mg/g, yet here, they would be in $\text{mg}/(\text{g-L}) \cdot \text{mL}$ (concentrations are in mg/L and V is in mL). The authors should change V here to L (or concentration to mg/mL) so that the units would cancel (and double check that their reported data is correctly in mg/g, and not $\text{mg}/(\text{g-L}) \cdot \text{mL}$)

The number of replicates are not clear in the methods. Were each condition in 3.3 tested in triplicates? There are standard errors shown in the figures so I think there were, but this should be explicit.

Page 3, line 13. 16S rDNA isn't a thing. The authors should change this to “16S rRNA gene” here and elsewhere.

Figure 1 – The figure legend needs to make clear what panel A, B, C, and D are showing (and x-axis labels on A and B are needed).

Decision letter (RSOS-181701.R0)

19-Nov-2018

Dear Dr Qiao:

Title: Bioimmobilization of lead by *Bacillus subtilis* X3 biomass isolated from lead mine soil under promotion of multiple adsorption mechanisms

Manuscript ID: RSOS-181701

Thank you for submitting the above manuscript to Royal Society Open Science. On behalf of the Editors and the Royal Society of Chemistry, I am pleased to inform you that your manuscript will be accepted for publication in Royal Society Open Science subject to minor revision in accordance with the referee suggestions. Please find the reviewers' comments at the end of this email.

The reviewers and handling editors have recommended publication, but also suggest some minor revisions to your manuscript. Therefore, I invite you to respond to the comments and revise your manuscript.

Because the schedule for publication is very tight, it is a condition of publication that you submit the revised version of your manuscript before 28-Nov-2018. Please note that the revision deadline will expire at 00.00am on this date. If you do not think you will be able to meet this date please let me know immediately.

Best wishes,
Dr Laura Smith
Publishing Editor, Journals

On behalf of the Subject Editor Professor Anthony Stace and the Associate Editor Dr Andrew Harned.

RSC Associate Editor:

Comments to the Author:

The referees have several relatively minor corrections that should be addressed by the authors in order to improve the readers' understanding of this work.

RSC Subject Editor:

Comments to the Author:

(There are no comments.)

Reviewer comments to Author:

Reviewer: 1

Comments to the Author(s)

In this paper, a lead-resistant bacterium, *Bacillus subtilis* X3, was used to prepare a lead bioadsorbent for immobilization and removal of lead in lead solution. Over the topic is interesting, however the paper is missing so much basic information. Major revision is needed if this paper can be accepted.

1. Page 3, line 24, The authors should explain why the adsorption capacity increased with the increase of pH, not just present a phenomenon.
2. Fig. 1(A) and (B), what do the abscissas represent?
3. Fig. 1(C), this figure should be revised and the title of abscissas should be revised to adsorbent concentration (g/L).
4. 4.3.3 XRD analysis, page 4, line 32-33, the authors stated that Pb^{2+} was transferred to $Pb_5(PO_4)_3$, $Pb_{10}(PO_4)_6(OH)_2$ and $Pb_5(PO_4)_2Cl$. However, there is no any analysis. You absolutely have to use citations to back up your statements.
5. The experimental conditions of Figure 1 should be provided in figure title.
6. The title of figure 3. XPS spectra of the lead shot precipitate and *B. subtilis*X3 biomass, (a) full spectrum; (b) Pb-4f spectrum; (c) C-1s spectrum. However, there is only one figure. Please explain.
7. The figures in page 9-11 and figures in page 12-16 are same.

Reviewer: 2

Comments to the Author(s)

This manuscript by Qiao, Zhang, and Luo is an overall well-written report on their study with lead bioimmobilization by *Bacillus*. The authors used several lines of evidence to show sorption of lead onto the biomass. There are only a few details in the manuscript that need to be edited/added in order to make the methods more clear.

Specific comments:

Page 1 Line 22 – the “subtilis” is not capitalized.

Page 2 Line 27 – What does ‘manually’ mean here with regards to genomic DNA extraction. Is there a protocol to reference? If not, some brief details on genomic DNA extraction need to be provided.

Page 2 Line 28 – What primers were used for the PCR of the 16S rRNA gene?

Page 2 line 43 – What is the form of lead here? Is it lead nitrate still? Be specific.

Page 2 line 59 – The q_e in the figures is in terms of mg/g, yet here, they would be in mg/(g-L)*mL (concentrations are in mg/L and V is in mL). The authors should change V here to L (or concentration to mg/mL) so that the units would cancel (and double check that their reported data is correctly in mg/g, and not mg/(g-L)*mL)

The number of replicates are not clear in the methods. Were each condition in 3.3 tested in triplicates? There are standard errors shown in the figures so I think there were, but this should be explicit.

Page 3, line 13. 16S rDNA isn't a thing. The authors should change this to "16S rRNA gene" here and elsewhere.

Figure 1 – The figure legend needs to make clear what panel A, B, C, and D are showing (and x-axis labels on A and B are needed).

Author's Response to Decision Letter for (RSOS-181701.R0)

See Appendix A.

Decision letter (RSOS-181701.R1)

03-Dec-2018

Dear Dr Qiao:

Title: Bioimmobilization of lead by *Bacillus subtilis* X3 biomass isolated from lead mine soil under promotion of multiple adsorption mechanisms

Manuscript ID: RSOS-181701.R1

Thank you for submitting the above manuscript to Royal Society Open Science. On behalf of the Editors and the Royal Society of Chemistry, I am pleased to inform you that your manuscript will be accepted for publication in Royal Society Open Science subject to minor revision in accordance with the referee suggestions. Please find the reviewers' comments at the end of this email.

The reviewers and handling editors have recommended publication, but also suggest some minor revisions to your manuscript. Therefore, I invite you to respond to the comments and revise your manuscript.

Because the schedule for publication is very tight, it is a condition of publication that you submit the revised version of your manuscript before 12-Dec-2018. Please note that the revision deadline will expire at 00.00am on this date. If you do not think you will be able to meet this date please let me know immediately.

Best wishes,

Dr Laura Smith
Publishing Editor, Journals

On behalf of the Subject Editor Professor Anthony Stace and the Associate Editor Dr Andrew Harned.

RSC Associate Editor

Comments to the Author:

The authors have done a good job addressing most of the concerns raised by the previous reviewers. However, there is still one item they should clarify.

One of the reviewers raised the following question:

"Comment: The title of figure 3. XPS spectra of the lead shot precipitate and *B. subtilis*X3 biomass, (a) full spectrum; (b) Pb-4f spectrum; (c) C-1s spectrum. However, there is only one figure. Please explain."

The author's response was:

"Response: We are sorry for the mistake. Here we only analyzed Pb-4f spectrum of the lead shot, so there is only one figure here."

The figure caption for Figure 3 still reads the same and strongly implies that there are multiple parts to this figure, but there is clearly only one part there. Please revise the Figure caption so it is more representative of what is there.

Reviewer comments to Author:

Author's Response to Decision Letter for (RSOS-181701.R1)

See Appendix B.

Decision letter (RSOS-181701.R2)

02-Jan-2019

Dear Dr Qiao:

Title: Bioimmobilization of lead by *Bacillus subtilis* X3 biomass isolated from lead mine soil under promotion of multiple adsorption mechanisms
Manuscript ID: RSOS-181701.R2

It is a pleasure to accept your manuscript in its current form for publication in Royal Society Open Science. The chemistry content of Royal Society Open Science is published in collaboration with the Royal Society of Chemistry.

On behalf of the Subject Editor Professor Anthony Stace and the Associate Editor Dr Andrew Harned.

RSC Associate Editor
Comments to the Author:
(There are no comments.)

Reviewer(s)' Comments to Author:

Appendix A

Dear Editor and Reviewers:

Thank you for your letter and for the reviewers' comments concerning our manuscript entitled "Bioimmobilization of lead by *Bacillus subtilis* X3 biomass isolated from lead mine soil under promotion of multiple adsorption mechanisms" (No.: RSOS-181701). Those comments are all valuable and very helpful for improving our paper, and they are also an important reference to our future researches. We have carefully considered the comments and have made corrections.

Reviewer # 1:

(1) Comment: Page 3, line 24, The authors should explain why the adsorption capacity increased with the increase of pH, not just present a phenomenon.

Response: We have added the explanation of the phenomenon in page3, Line 3-8 (5. Discussion). That is: "Ren et al. (2015) pointed out that pH is one of the most important parameters affecting the solubility of metal ions and functional groups on the cell wall of biomass. A functional group such as a carboxylate on the bacterial cell wall is protonated below 4.0, and Pb(II) is weak in competition with hydrogen ions at the adsorption site on the surface of the cell. Therefore, the

lower Pb(II) absorption capacity was caused by more hydrogen ions at a pH of 3.0 to 4.0 ”.

Reference:

Ren, G., Jin, Y., Zhang, C., Gu, H., Qu, J. 2015 Characteristics of *Bacillus sp.* PZ-1 and its biosorption to Pb(II). *Ecotoxicol. Environ. Saf.* 117, 141-148

(2) Comment: Fig. 1(A) and (B), what do the abscissas represent?

Response: The abscissas in Fig.1 (A) is “pH”, and that in Fig.1(B) is “Time (min)”. We have added the abscissas in the figure.

(3) Comment: Fig. 1(C), this figure should be revised and the title of abscissas should be revised to adsorbent concentration (g/L).

Response: We have revised the abscissas in Fig.1(C) according to the reviewer’s comment. The abscissas “ adsorbent quality(g)” has been replaced by “adsorbent concentration (g/L)”.

(4) Comment: 4.3.3 XRD analysis, page 4, line 32-33, the authors stated that Pb²⁺ was transferred to Pb₅(PO₄)₃, Pb₁₀(PO₄)₆(OH)₂ and Pb₅(PO₄)₂Cl. However, there is no any analysis. You absolutely have to use citations to back up your statements.

Response: The MID Jade software can calculate the chemical structure to obtain the molecular according to the XRD spectra of lead shot. Previous studies had similar statements (Levinson et al.; Mire et al.), which was mentioned in our manuscript(Page 5,Line 9)

References:

Levinson, H. S., Mahler, I. 1998 Phosphatase activity and lead resistance in *Citrobacter freundii* and *Staphylococcus aureus*. FEMS Microbiol. Lett. 161, 135-138. (doi.org/10.1111/j.1574-6968.1998.tb12939.x)

Mire, C. E., Tourjee, J. A., O'Brien, W. F., Ramanujachary, K. V.,

Hecht, G. B. 2004 Lead Precipitation by *Vibrio harveyi*: Evidence for Novel Quorum-Sensing Interactions. Appl. Environ. Microbiol. 70, 855-864.

(5) Comment: The experimental conditions of Figure 1 should be provided in figure title.

Response: We have added the experimental conditions (in a shaker at 25°C and 150 rpm for half an hour) to the caption of Fig.1.

(6) **Comment:** The title of figure 3. XPS spectra of the lead shot precipitate and *B. subtilis*X3 biomass, (a) full spectrum; (b) Pb-4f spectrum; (c) C-1s spectrum. However, there is only one figure. Please explain.

Response: We are sorry for the mistake. Here we only analyzed Pb-4f spectrum of the lead shot, so there is only one figure here.

(7) **Comment:** The figures in page 9-11 and figures in page 12-16 are same.

Response: We have deleted the same figures.

Reviewer # 1:

(1) Comment: Page 1 Line 22 – the “subtilis” is not capitalized.

Response: We have revised the “*Bacillus subtilis* X3” to “*Bacillus subtilis* X3”.

(2) Comment: Page 2 Line 27 – What does ‘manually’ mean here with regards to genomic DNA extraction. Is there a protocol to reference? If not, some brief details on genomic DNA extraction need to be provided.

Response: Genomic DNA of strain X3 was extracted using Bacterial Genomic DNA Extraction Kit (Sangon, China). We extracted the genomic DNA according to the protocol of the Kit, which is the conventional operation.

(3) Comment: Page 2 Line 28 – What primers were used for the PCR of the 16S rRNA gene?

Response: We have added the primers in our manuscript:
f1 (5'-AGTTTGATCMTGGCTCAG-3') and r1
(5'-GGTTACCTTGTTACGACTT-3')

(4) Comment: Page 2 line 43 – What is the form of lead here? Is it lead nitrate still? Be specific.

Response: The solution contained $\text{Pb}(\text{NO}_3)_2$ (200, 400, 600, 800, 1000, 1200, 1400 mg L^{-1} of Pb^{2+}). We have revised the sentence.

(5) Comment: Page 2 line 59 – The q_e in the figures is in terms of mg/g, yet here, they would be in $\text{mg}/(\text{g-L})\cdot\text{mL}$ (concentrations are in mg/L and V is in mL). The authors should change V here to L (or concentration to mg/mL) so that the units would cancel (and double check that their reported data is correctly in mg/g, and not $\text{mg}/(\text{g-L})\cdot\text{mL}$)

Response: We have revised the mistake. That is: "... C_0 and C_e were the initial and final concentrations of lead (mg/L), respectively and V and M were the volume of solution (L) and the weight of the biomass (g), respectively".

(6) Comment: The number of replicates are not clear in the methods. Were each condition in 3.3 tested in triplicates? There are standard errors shown in the figures so I think there were, but this should be explicit.

Response: Each experiment was performed with three biological and technical replicates. The 95% confidence interval ($P < 0.05$) was set as the significance threshold. We have added the sentence in our manuscript.

(7) Comment: Page 3, line 13. 16S rDNA isn't a thing. The authors should change this to "16S rRNA gene" here and elsewhere.

Response: We have corrected all the mistake according to the reviewer's comment.

(8) Comment: Figure 1 – The figure legend needs to make clear what panel A, B, C, and D are showing (and x-axis labels on A and B are needed).

Response: We have corrected Fig.1 according to the reviewer's comment.

Appendix B

Dear Editor and Reviewers:

Thank you for your letter and for the reviewers' comments concerning our manuscript entitled "Bioimmobilization of lead by *Bacillus subtilis* X3 biomass isolated from lead mine soil under promotion of multiple adsorption mechanisms" (No.: RSOS-181701.R1). Those comments are all valuable and very helpful for improving our paper, and they are also an important reference to our future researches. We have carefully considered the comments and have made corrections.

Comment: The authors have done a good job addressing most of the concerns raised by the previous reviewers. However, there is still one item they should clarify.

One of the reviewers raised the following question:

"Comment: The title of figure 3. XPS spectra of the lead shot precipitate and *B. subtilis*X3 biomass, (a) full spectrum; (b) Pb-4f spectrum; (c) C-1s spectrum. However, there is only one figure. Please explain."

The author's response was:

"Response: We are sorry for the mistake. Here we only analyzed Pb-4f spectrum of the lead shot, so there is only one figure here."

The figure caption for Figure 3 still reads the same and strongly implies that there are multiple parts to this figure, but there is clearly only one part there. Please revise the Figure caption so it is more representative of what is there.

Response: We are sorry for the mistake. We have checked Fig.3 and its caption carefully. Then we added a part in Fig.3 and revised the caption for Fig.3. The caption is: “XPS spectra (Pb-4f spectrum) of the lead shot precipitate and *B. subtilis* X3 biomass, (a) Pb-4f spectrum; (b) C-1s spectrum”